# Assessment of breath volatile organic compounds in acute cardiorespiratory breathlessness: a protocol describing a prospective real-world observational study

Wadah Ibrahim,[1,2] Michael Wilde,[3] Rebecca Cordell,[3] Dahlia Salman,[4] Dorota Ruszkiewicz,[4] Luke Bryant,[3] Matthew Richardson,[1] Robert C Free,[1] Bo Zhao,[1] Ahmed Yousuf,[1,2] Christobelle White,[1,2] Richard Russell,[1,2] Sheila Jones,[2] Bharti Patel,[2] Asia Awal,[2] Rachael Phillips,[5] Graham Fowkes,[5] Teresa McNally,[6] Clare Foxon,[6] Hetan Bhatt,[1] Rosa Peltrini,[1] Amisha Singapuri,[1] Beverley Hargadon,[1] Toru Suzuki,[7,8] Leong L Ng,[7,8] Erol Gaillard,[6] Caroline Beardsmore,[6] Kimuli Ryanna,[2] Hitesh Pandya,[9] Tim Coates,[1,10] Paul S Monks,[3] Neil Greening,[1] Christopher E Brightling,[1] Paul Thomas,[4] Salman Siddiqui[1]

For numbered affiliations see end of article.

**Correspondence to**
Professor Salman Siddiqui; ss338@leicester.ac.uk

## ABSTRACT

**Introduction** Patients presenting with acute undifferentiated breathlessness are commonly encountered in admissions units across the UK. Existing blood biomarkers have clinical utility in distinguishing patients with single organ pathologies but have poor discriminatory power in multifactorial presentations. Evaluation of volatile organic compounds (VOCs) in exhaled breath offers the potential to develop biomarkers of disease states that underpin acute cardiorespiratory breathlessness, owing to their proximity to the cardiorespiratory system. To date, there has been no systematic evaluation of VOC in acute cardiorespiratory breathlessness. The proposed study will seek to use both offline and online VOC technologies to evaluate the predictive value of VOC in identifying common conditions that present with acute cardiorespiratory breathlessness.

**Methods and analysis** A prospective real-world observational study carried out across three acute admissions units within Leicestershire. Participants with self-reported acute breathlessness, with a confirmed primary diagnosis of either acute heart failure, community-acquired pneumonia and acute exacerbation of asthma or chronic obstructive pulmonary disease will be recruited within 24 hours of admission. Additionally, school-age children admitted with severe asthma will be evaluated. All participants will undergo breath sampling on admission and on recovery following discharge. A range of online technologies including: proton transfer reaction mass spectrometry, gas chromatography ion mobility spectrometry, atmospheric pressure chemical ionisation-mass spectrometry and offline technologies including gas chromatography mass spectroscopy and comprehensive two-dimensional gas chromatography-mass spectrometry will be used for VOC discovery and replication. For offline technologies, a standardised CE-marked breath sampling device (ReCIVA) will be used. All recruited participants will be characterised using existing blood biomarkers including C reactive protein, brain-derived natriuretic peptide, troponin-I and blood eosinophil levels and further evaluated using a range of standardised questionnaires, lung function testing, sputum cell counts and other diagnostic tests pertinent to acute disease.

**Ethics and dissemination** The National Research Ethics Service Committee East Midlands has approved the study protocol (REC number: 16/LO/1747). Integrated Research Approval System (IRAS) 198921. Findings will be presented at academic conferences and published in peer-reviewed scientific journals. Dissemination will be facilitated via a partnership with the East Midlands

## Strengths and limitations of this study

► A pragmatic real-world, prospective, observational study across three admission units that focuses on the systematic discovery and replication of volatile organic compound (VOC) in acutely breathless patients using both online and offline technologies.

► The proposed study is the largest of its kind in acute disease to characterise VOC with a range of additional assessments that will build a comprehensive phenotype of acute cardiorespiratory exacerbations.

► The proposed study will build an infrastructure for research and subsequent evaluation of VOC in interventional trials within acute cardiorespiratory exacerbations.

► Prior acute treatment exposure will need to be accounted for when evaluating potential discriminative biomarkers.

► VOC technologies are not currently suited for deployment in patients that are of high clinical acuity.

Academic Health Sciences Network and via interaction with all UK-funded Medical Research Council and Engineering and Physical Sciences Research Council molecular pathology nodes.

**Trial registration number** NCT03672994.

## INTRODUCTION

Breathlessness is a common symptom of cardiorespiratory illnesses that has a significant direct impact on patients' well-being as well as a substantial economic burden on healthcare systems.[1] Although its aetiologies can be variable, exacerbations of common complex chronic cardiorespiratory conditions account for approximately 70% of acute presentations with breathlessness, namely exacerbations of asthma and chronic obstructive pulmonary disease (COPD), acute heart failure and community acquired pneumonia.[2] Moreover, moderate and severe breathlessness is significantly associated with all-cause, cardiovascular and COPD mortality.[3] As a consequence, symptomatic breathlessness warrants rapid evaluation and targeted diagnostics at presentation.

Diagnostic evaluation of acute breathlessness is heavily reliant on blood-based biomarkers for example, C reactive protein (CRP), brain-derived natriuretic peptide (BNP), troponin and on occasions blood eosinophil levels. These biomarkers have clinical utility primarily in patients with single pathologies but have poor discriminatory power in patients with multifactorial presentations of acute breathlessness.[4] There is therefore an unmet need for the development of sensitive and specific biomarkers that differentiate acute breathlessness from its recovery and the common cardiorespiratory conditions that present with acute breathlessness.

CRP plays an important role in diagnosing breathlessness caused by an underlying bacterial pneumonia,[5] as well as predicting mortality in patients with chronic obstructive pulmonary disease (COPD).[6] BNP is routinely used in acute settings to support the diagnosis of acute heart failure.[7] The European Society of Cardiology recommends BNP threshold values of <100 pg/mL to rule out acute congestive cardiac failure and values >500 pg/mL as diagnostic of acute exacerbations of heart failure.[8]

The role of peripheral blood eosinophil count in airway inflammation was poorly understood up until the second half of the 19th century when Paul Ehrlich, a German physician and Nobel prize winner, introduced eosin in his technique for white cell differentiation in 1879.[9] Considerable advances in the field of airway inflammation and the role of eosinophils have taken place since.[10–12] More recently, Bafadhel et al[13] suggested that peripheral blood eosinophil count can be used to direct corticosteroid therapy during COPD exacerbations in single-centre study.

Currently, blood biomarkers together with clinical, physiological and imaging parameters are used in diagnosing the cause of acute breathlessness. Blood biomarkers may be less specific as they originate far from the target organs of interest (the heart and the lungs in

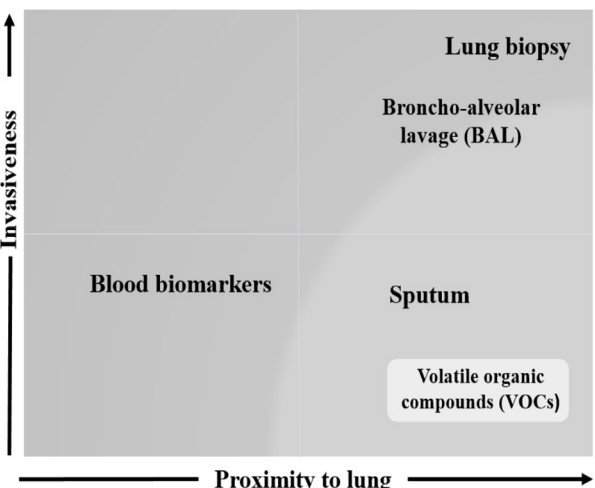

**Figure 1** Relationship between lung proximity and degree of invasiveness of different lung matrices. The figure plots the level of invasiveness of various lung matrices in relation to their proximity to the lung. Given their pathological relevance, the degree of invasiveness of bronchoalveolar lavage (BAL) and lung biopsy makes them less favourable in diagnosing respiratory diseases.

cardiorespiratory disease). Sputum, although potentially a more definitive lung-specific matrix, is comparatively difficult to obtain particularly in acutely unwell patients, limiting its use in acute disease and highlighting the need for better biomarkers. Ideally, these biomarkers would have the following characteristics: (1) they would originate from the target organ of interest, (2) they would significantly add value to conventional risk scoring and diagnostic algorithms in acute breathlessness, (3) they would be minimally invasive and suitable for rapid point of care diagnosis in emergency rooms and acute admissions units and (4) they would have diagnostic value in patients with multifactorial acute breathlessness.

Exhaled breath contains thousands of volatile organic compounds (VOCs) that reflect biological processes occurring in the host both locally in the airways and systematically offering the potential to develop more effective biomarkers in acutely breathless patients (figure 1).

The proposed programme of research will use a combination of offline and online technologies to identify and evaluate the diagnostic and prognostic value of VOC in patients with acute cardiorespiratory-related breathlessness (figure 2).

Exhaled breath analysis of volatile chemicals offers the potential to develop more effective biomarkers in acutely breathless patients. The use of breath analysis for disease diagnostics dates back to ancient Greeks where physicians used exhaled breath to diagnose different diseases. Breath odours allow correct associations to certain diseases. For example, the sweet smell of diabetic ketoacidosis, the fishy smell of breath associated to liver illness, the urine-like odour of kidney disease and the smell of the breath

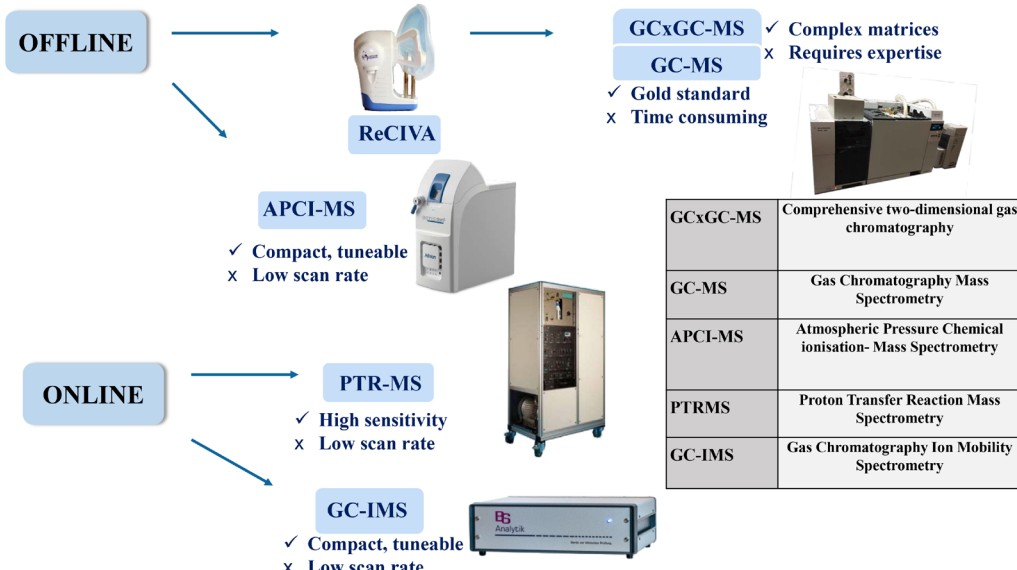

**Figure 2** Multi-instrument use in breath sampling. Figure illustrates the various combinations of offline and online devices used in breath sampling and the relevant pros and cons. Offline and online technologies are used for the discovery and validation phases of the study, respectively.

of patients with lung abscesses caused by the proliferation of anaerobic bacteria.[14–17]

More recently, exhaled breath analysis has demonstrated early proof of concept in the diagnosis of acute heart failure, and ventilator associated pneumonia.[18] The validity of breath analysis has also been demonstrated in breathless children.[19] This population is likely to prefer breath-based tests, as these are minimally invasive. Importantly, a variety of point-of-care sensors are now available to evaluate potential exhaled breath biomarkers in emergency care settings.

A study by Van Berkel et al[20] demonstrated the ability to distinguish COPD subjects from controls solely based on the presence of VOCs in breath, suggesting that analysis of VOC might be highly relevant for diagnosis of COPD. This established the basis of further studies of VOC in COPD[21–25] recommending larger studies for validation.

Several other studies found that VOC profiling in diagnosing asthma is potentially feasible.[26–32] This, however, has been done in relatively small numbers in stable disease.

Despite the novelty of non-invasive sampling technology and the growing interest in exhaled breath analysis, there remains a disappointing level of comparability across studies due to the lack of standardisation and appropriate data analysis methods. A recent systemic review by Anders Christiansen et al[33] compared 11 publications reporting very heterogeneous designs, methods, patient group sizes, data analytics and, consequently, quite varying results.

To our knowledge, no other large studies exploring the use of breath biomarkers in profiling acute breathlessness have been completed. Several studies have explored the use of electronic nose (eNose) in stable disease with good discriminatory power in COPD,[34] pneumonia[35] and heart failure[36] with relatively small sample size. While eNose has now been widely used in detecting various VOC patterns, gas chromatography mass spectroscopy (GC-MS), a largely validated methodology, remains the gold standard technique for detecting VOCs in exhaled breath. The focus of the current research study will be to evaluate acutely breathless cardiorespiratory patients using a combination of 'discovery' and near-patient care breath sampling technologies.

Medical Research Council (MRC) and Engineering and Physical Sciences Research Council have commissioned a series of molecular pathology nodes aimed at developing molecular signatures relevant to disease diagnosis and progression. This was triggered by the clear need for alliance between academic institutions, industry and National Health Service partners to enhance the benefits of stratified medicine for patients.[37]

University of Leicester and Loughborough University were awarded a joint molecular pathology node East Midlands Breathomics Pathology Node (EMBER), which this study forms a key part of.

## METHODS AND ANALYSIS
### Study design
A prospective real-world observational study across three acute admissions units within Leicestershire (two adult admissions units and one children's assessment unit). The acute units routinely assess and treat cardiorespiratory admissions due to breathlessness in adults and children.

Participants with self-reported acute breathlessness, either requiring admission or a change in baseline treatment, will be screened for the study. Informed consent will be obtained in all participants following a clinical review by a senior decision maker within 24 hours of acute admission (figure 3).

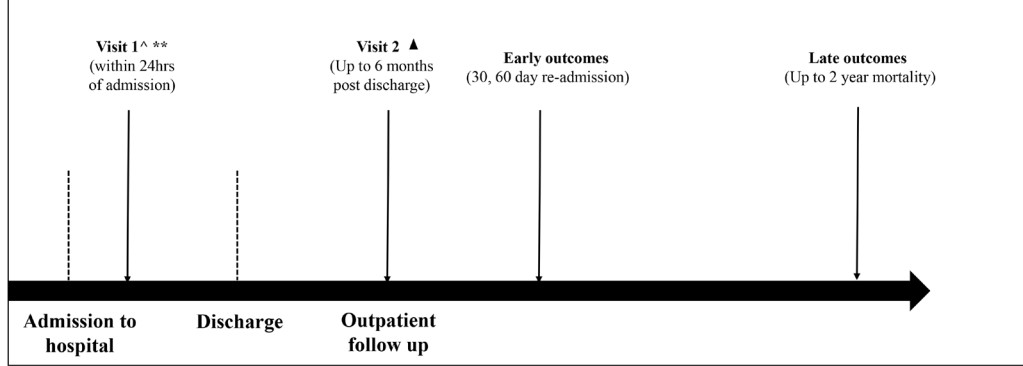

^    Following senior decision maker review
**   Breath sampling (ReCIVA-GC-MS, ReCIVA- GCxGC-MS, GC-IMS, APCI-MS)
▲    Breath sampling (ReCIVA-GC-MS, ReCIVA- GCxGC-MS, PTR-MS, APCI-MS)
----  Following University Hospitals of Leicester streaming and clinical care pathways

**Figure 3**   Study flow chart. Figure outlines the patient journey from admission through to discharge and follow-up. Participants with self-reported acute breathlessness presenting to University Hospitals of Leicester are recruited within 24 hours following a senior decision maker review. Breath sampling is carried out on the first visit, at recruitment, and the second visit, up to 6 months after discharge. Patients are admitted through the standard operational emergency medical streaming and care pathways at the University Hospitals of Leicester National Health Service Trust. Early outcomes (hospital readmission) are measured at 30 days and 60 days and late outcomes (mortality) including respiratory and all-cause mortality are measured at 2 years. Assessments carried out at each time point are summarised in table 1. APCI-MS, atmospheric pressure chemical ionisation-mass spectrometry; GC-IMS, gas chromatography ion mobility spectrometry; GC-MS, gas chromatography mass spectroscopy; GC×GC-MS, two-dimensional gas chromatography-mass spectrometry.

## Objectives
### Primary objective
► To evaluate the sensitivity, specificity, positive and negative predictive values of exhaled breath VOC biomarkers to differentiate acute breathlessness in cardiorespiratory patients.

### Secondary objectives
► To replicate selected breath VOC biomarkers identified in acute breathlessness.
► To discover and replicate breath VOC biomarkers that differentiate the common cardiorespiratory conditions that cause acute breathlessness, specifically: (1) acute heart failure, (2) community-acquired pneumonia, (3) adult exacerbations of asthma and chronic obstructive pulmonary disease (COPD) and age-matched adults that do not have cardiorespiratory disease or breathlessness.
► To quantify the level of clinical uncertainty in the primary diagnosis using a 100 mm visual analogue scale (VAS) and independent clinical adjudication of case notes blinded to the following blood biomarkers: (1) CRP, (2) BNP, (3) troponin-I and (4) blood eosinophils but not clinical history and acute presentation nor chest X-ray imaging. Potential discriminatory breath VOC biomarkers will be adjusted for clinical uncertainty in statistical models.
► To identify and replicate exhaled breath VOC biomarkers in school-age children treated in hospital for severe asthma attacks and compare these with age-matched healthy controls.

### Exploratory endpoints (where applicable)
► To evaluate the dynamic profile of selected breath VOC between the acute state and the recovery state postexacerbation.
► To evaluate the relationship between exhaled VOC biomarkers and clinical outcomes including: (1) hospital readmission at 30 days and 60 days postevent and (2) all-cause mortality over a 2-year period postadmission.
► To evaluate the relationship between breath VOC biomarkers and functional measures, for example, physical performance and activity.
► To explore potential breath VOC biomarkers of multi-factorial acute breathlessness.
► To evaluate the relationship between diet, lifestyle and environment on breath VOC biomarkers.

### Sample size estimation
Preliminary data were used to conduct sample size estimates from a cohort of acutely breathless patients admitted to acute admissions units over a 6-month period (February 2017–August 2017). One hundred and twelve adult participants (asthma: 46, community-acquired pneumonia: 26 and COPD: 22) and 18 healthy controls were used for the analysis.

A panel of 10 prespecified aldehydes, based on literature search,[31 38 39] were extracted from breath using GC-MS. The aldehydes were normalised to a common internal standard and were not background-subtracted.

A closed formula from Hsieh *et al*,[40] relating sample size to observable effect size, was used to calculate sample size

from logistic regression models of the ten aldehydes with acute breathlessness as the outcome measure. The sample size estimates are also relevant to acute class comparisons versus the sum of other acute classes.

Based on the sample size estimates, we would have an 80% power, with a type 1 error rate of 5%, to detect an OR of association of 1.2 between two disease classes with 55 patients per class. Given the fact that study seeks to discover and replicate breath VOC among five adult disease classes (community-acquired pneumonia, heart failure, COPD, asthma and healthy aged-matched subjects), we would require 110 adult patients per class—550 patients across the programme to achieve these aims.

The closed formulae by Hajian-Tilaki et al[41] were also used to understand the discriminatory power that the samples sizes above would provide with respect to biomarker sensitivity and specificity; the following assumptions were made:

► That a sensitivity of 80% with a precision of 5% would provide a useful biomarker capable of 'ruling out' an acute class. The same target was applied to specificity.
► We assume a prevalence of acute breathlessness of 80% as the recruitment campaign uses acute breathlessness as the initial stratification tool for recruitment and 1:5 patients recruited will be non-breathless healthy controls.
► We aim to balance group sizes across classes equally.

For a type 1 error rate of 0.05 and a 95% CI:

$N_{sensitivity}$=307.
$N_{specificity}$=1230.

For a type 1 error rate of 0.05 and a 90% CI:

$N_{sensitivity}$=218.
$N_{specificity}$=871.

For a type 1 error rate of 0.05 and an 85% CI:

$N_{sensitivity}$=166.
$N_{specificity}$=664.

For a type 1 error rate of 0.05 and an 80% CI:

$N_{sensitivity}$=131.
$N_{specificity}$=524.

Therefore, we are powered to identify sensitive biomarkers (≥80%) of acute breathlessness with a maximum marginal error in the estimate for sensitivity not exceeding 5% with 95% confidence. Similarly, we are powered to identify specific biomarkers (≥80%) of acute breathlessness with a maximum marginal error in the estimate for specificity not exceeding 5% with 80% confidence.

For the primary analysis, the outcome will be treated as a nominal variable with levels: (1) acute heart failure, (2) community-acquired pneumonia; (3) adult exacerbations of asthma and COPD; and (4) acute exacerbations in school-age children treated in hospital for severe asthma attacks.

The relationship between the primary outcome and the exhaled breath VOC biomarkers will be modelled using multinomial logistic regression. In addition to metabolomics markers, the following independent variables will be included in the model: clinical uncertainty score on a 100 mm VAS scale, age and a validated comorbidity score (the Charlson comorbidity score).[42 43]

Receiver operator analyses will be used to generate reciver operating characteristic curves for individual and multiple panels VOC predictors in the primary analysis.

To understand the dynamic profile of breath biomarkers during (1) the acute state and (2) in the chronic state up to 6-month postexacerbation, a repeated measures model with a random intercept and random effect for time will be fitted; the random effects will be fitted for each patient. For the repeated measures mixed model, an unstructured covariance will be assumed. To evaluate the relationship between breath biomarkers and hospital readmission at 30 and 60 days, Cox proportional hazards and frailty models will be used.[44] Analysis of multivariate survival data, competing risk models and joint models will be fitted.[45] Relationship between death and breath biomarkers will be evaluated using a logistic regression model. Changes in outcome measures will be measured appropriately for each variable (eg, paired t-test, Mann-Whitney and repeated measures analysis). Tables of descriptive statistics will be compiled for all key variables.

All analysis will be performed using R V.3.5.0 (https://www.r-project.org/).

### Discovery and replication studies

Specific indicator conditions have been selected for targeted recruitment according to their high prevalence and unmet need, their high morbidity and mortality and the need to develop better diagnostic and prognostic algorithms in acute care pathways.

The indicator diagnoses of interest are: (1) exacerbations of adult asthma and COPD, (2) community-acquired pneumonia, (3) acute heart failure; and (4) exacerbation in school age children treated in hospital for severe asthma attacks.

Patient-level clinicopathological and outcome data (spanning the entire acute pathway) will be collected in parallel to breath sampling. In addition, breath samples will be acquired in the stable state post exacerbation (figure 3).

Age-matched healthy volunteers will be recruited where possible at separate visits. For the purposes of this study, healthy volunteers will be defined as participants who have no prior history of asthma, COPD, heart failure and have not been admitted to hospital with community acquired pneumonia within 6 weeks of the baseline study visit. For acute admission, the study team will approach the spouse, parent or sibling of the index case and seek informed consent for study assessments. All healthy subjects will undergo two assessments separated by a duration of 8–16 weeks to match the acute and recovery time points elapsed in their index case/partner/spouse/sibling/child. Additional healthy volunteers will be identified from local recruitment databases and via advertising

**Table 1** Table summarising recruitment targets for both adult and paediatric groups. Total combined sample size of the discovery and replication phases = 700 participants.

| Disease category | Discovery | Replication |
|---|---|---|
| Acute adult asthma | 55 | 55 |
| Acute chronic obstructive pulmonary disease | 55 | 55 |
| Acute heart failure | 55 | 55 |
| Community-acquired pneumonia | 55 | 55 |
| Adult healthy volunteers | 55 | 55 |
| Acute paediatrics asthma | 50 | 25 |
| Paediatrics healthy volunteers | 50 | 25 |
| **Total sample** | 375 | 325 |

## Discovery phase (project months 1–24)

The aim of the discovery phase is to discover putative discriminatory breath VOC, using both offline and online technologies.

Preplanned recruitment of acutely breathless patients will be enriched into the following disease strata following senior clinical decision maker assessment and within 24 hours of acute admission.

Acute adult heart failure (n=55), adult community-acquired pneumonia (n=55), adult exacerbations of asthma (n=55) and COPD (n=55) in addition to acute severe asthma attacks in school-age children (n=50).

Additional age-matched healthy volunteers (n=55 adults and 50 children) will be identified as a non-disease reference group (table 1).

## Replication phase (years 3–4)

The aim of the replication phase is to replicate putative discriminatory breath VOC/VOC signatures identified in the discovery phase.

Similar to the discovery phase, recruitment of acutely breathless patients will be enriched into the following disease strata following senior clinical decision maker assessment and within 24 hours of acute admission.

Acute adult heart failure (n=55), adult community acquired pneumonia (n=55), adult exacerbations of asthma (n=55) and COPD (n=55) in addition to acute severe asthma attacks in school age children (n=25) (table 1).

Additional age-matched healthy volunteers (n=55 adults and 25 children) will be identified as a non-disease reference group.

## Schedule of assessments

A schedule of acute assessments is outlined below and aligns to the movement of acute patients through the clinical care pathway and the overall aim of developing a complete phenotypic picture of acutely breathless patients.

## Defining acute breathlessness

At presentation (within 24 hours of admission) to one of three acute admissions units, potentially eligible patients will be identified following confirmation of acute breathlessness, identified as: (1) patient-defined acute breathlessness and/or (2) 1 unit increase above patient reported baseline in the extended MRC (eMRC) dyspnoea score[46 47] and at least one of the indicator diagnoses identified as the primary clinical diagnosis by a senior clinical decision maker. eMRC will be completed by all patients and healthy volunteers at each research visit.

## Informed consent

Patients meeting the prespecified definition of acute breathlessness will be approached for informed consent to the breath VOC biomarker study. Only patients that are eligible to give full written informed consent will be recruited.

## Collection of blood-based pathology markers

Collection of the blood biomarkers CRP, BNP, troponin-I and blood eosinophil count will be performed both acutely and following recovery, when not taken as part of clinical care pathway. These are currently used in profiling acutely breathless patients in clinical practice (table 2).

## Breath VOC sampling

Offline breath sampling using GC-MS and comprehensive GC×GC-MS coupled with a standardised and CE-marked breath sampler ReCIVA[48] will be performed. Gas chromatography is considered a gold standard technique in detecting VOCs and as such its sampling will be prioritised. Additionally, the following online technologies, proton transfer reaction mass spectroscopy (PTR-MS), gas chromatography ion mobility spectrometry (GC-IMS) and atmospheric pressure chemical ionisation-mass spectrometry (APCI-MS) will be evaluated according to the sampling strategy outlined in figure 3 and table 3.

## Collection of additional samples for future biomarker campaigns

Collection of additional biomarkers for future biomarker discovery campaigns including: (1) a urine sample, (2) blood samples, up to 85 mL for DNA, RNA, plasma and serum and peripheral blood cell flow cytometry in selected subjects and (3) spontaneous or induced sputum samples (plugs and supernatants) will be carried out (table 3).

All samples will be collected at time points 1 and 2 (figure 3). The additional samples will be used for future omics analyses, these may include detailed analysis of the metagenome in sputum and proteomics applied to urine and serum samples.

## Physiological characterisation

Physiological measures of lung function will be performed in acutely ill participants and at recovery including: (1) handheld forced oscillation technique: an easily accessible measure of lung function. Patients favour this to spirometry as it is effort independent, unlike spirometry, and requires less than a minute of

**Table 2** Type of analyser and methodology used for blood biomarker calculation

| Test | Analyser/method | Lower limit of detection | Upper limit of detection |
|---|---|---|---|
| C reactive protein | Siemens Advia Chemistry XPT, Polyethylene Glycol enhanced immunoturbidimetric assay. Siemens Advia 1800, Polyethylene Glycol enhanced immunoturbidimetric assay. | 5 mg/L | Diluted to result |
| B-type natriuretic peptide | Siemens Advia Centaur XPT, two-site sandwich immunoassay using direct chemiluminescent technology. | 2.0 pg/mL | 1445 pg/mL |
| Troponin-I | Abbott Architect i2000SR, three-site sandwich immunoassay using direct chemiluminescent technology (CMIA). | 5.0 ng/L | 50 000 ng/L |

The table outlines analyser make, methodology, upper and lower limits of detection as per the University Hospitals of Leicester National Health Service Foundation trust laboratory guidelines.

quiet tidal breathing to obtain triplicate high-quality measurements,[49] This will be completed using Tremoflo, Thorasys Thoracic Medical Systems Inc. (2) Fractional exhaled nitric oxide: a measure of airway inflammation in patients with asthma.[50 51] This instrument used for this will be NIOX VERO, registered trademark of Circassia AB (PP-VERO-UK-0022-V.1.0). (3) Echocardiography: two-dimensional transthoracic echocardiography will be performed in heart failure and COPD patients using an iE 33 system with S5-1 transducer (frequency transmitted 1.7 MHz, received 3.4 MHz; Philips Medical Systems, Best, The Netherlands). Standard techniques as per American Society of Echocardiography guidelines (ASE)[52] will be used to acquire two-dimensional, colour and Doppler images in conventional parasternal long-axis, short-axis and apical 4-chamber, 2-chamber and 3-chamber views. Left ventricular ejection fraction will be calculated using the biplane method of discs formula (Simpson's rule) to derive left ventricular volume indices.

All participants are encouraged to report any testing-related discomfort or concerns to the research team to terminate the sampling process.

### Recovery follow-up
► Patient recovery will be defined as:
  i. Patient-reported recovery from the acute exacerbation spell and back to their baseline-extended MRC score or clinician-defined recovery from the acute exacerbation spell.
  ii. At least 6 weeks postexacerbation event (up to 6 months).

Patients who readmit to hospital between visits 1 and 2 can have additional visit 1 assessments. Visit 2 will be taken as recovery following the subsequent admission. If a patient is admitted to hospital after visit 2, then they will be eligible to be recruited as a new study participant.

The schedule of assessments at the recovery visit is outlined in table 3.

### Clinical adjudication
In an effort to reduce data variability and minimise bias, an independent panel consisting of two senior acute clinicians (SS and NG) will review all pertinent clinical and diagnostic source documentation while blinded to admission blood biomarkers and clinical diagnosis.

All acutely breathless adult patients' notes will be sequentially divided for adjudication. The panel will independently determine the primary diagnosis of highest probability from a list of the four potential acute indicator diagnoses and mark their level of clinical certainty on a 100 mm VAS. The panel members will be able to review imaging, ECGs and other relevant information but not admission blood-based pathology tests.

In a subset of patients, adjudication will be validated by separate panel member to ensure between observer agreement using Bland-Altman analysis and inter-rater agreement of the primary diagnosis using Kohen's kappa via repeated evaluation of a subset of cases (see statistical methods outlined).

### Clinical informatics
Clinical data collection will be undertaken using a securely hosted bespoke database system (ADD) developed within the National Institute for Health Research (NIHR) Leicester Biomedical Research Centre – Respiratory (BRC). The system links acute admission episodes to hospital pathology records; historical respiratory physiology tests; and demographic information. The system provides functionality to validate data entry, manually verify records and highlight incomplete records. A custom VOC 'module' has been be created to support data collection within the study visits (1 and 2) and standardise diagnoses and medications through the use of clinical ontologies as well as linking hospital records/tests to patient visits.

Non-ADD-based clinical data (eg, hospital admissions, readmissions and mortality) will be extracted from the hospital data warehouse using identifiable patient

**Table 3** Summary of baseline and follow-up assessments. The table summarises key assessments carried out at different time points during the study. The participants may undertake any combination of the investigations listed at any of these time points

| Time point | Chronic obstructive pulmonary disease | | Asthma | | Pneumonia | | Heart failure | | Healthy | | Paediatrics | |
|---|---|---|---|---|---|---|---|---|---|---|---|---|
| | 1 | 2 | 1 | 2 | 1 | 2 | 1 | 2 | 1 | 2 | 1 | 2 |
| Written informed consent | × | | × | | × | | × | | × | | × | |
| **Volatile organic compound sampling** | | | | | | | | | | | | |
| ReCIVA – gas chromatography and mass spectrometry | × | × | × | × | × | × | × | × | × | × | × | × |
| ReCIVA comprehensive two-dimensional gas chromatography | × | × | × | × | × | × | × | × | × | × | × | × |
| Atmospheric pressure chemical ionisation mass spectrometry | × | × | × | × | × | × | × | × | × | × | × | × |
| Proton transfer reaction mass spectrometry | | × | | × | | × | | × | | × | | × |
| Gas chromatography ion mobility spectrometry | × | | × | | × | | × | | | | × | × |
| **Pathology blood tests** | | | | | | | | | | | | |
| Full blood count (including differential cell count) | × | × | × | × | × | × | × | × | × | × | × | × |
| Brain natriuretic peptide (pg/mL) | × | × | × | | × | | × | × | × | | | |
| Troponin-I (ng/L) | × | × | × | | × | | × | | × | | | |
| C reactive protein (mg/L) | × | × | × | × | × | × | × | × | × | × | × | × |
| **Lung function tests** | | | | | | | | | | | | |
| Hand held forced oscillation technique | × | × | × | × | × | × | × | × | × | × | | |
| Fractional exhaled nitric oxide – flow rate 50 (mL/s) | | | × | × | | | × | × | × | × | × | × |
| Spontaneous sputum sample | × | × | × | × | × | × | × | × | × | × | | |
| Biobanking (urine, serum, plasma. sputum supernatants and plugs) | × | × | × | × | × | × | × | × | × | × | × | × |
| Transthoracic echocardiography | × | | | | | | × | | | | | |

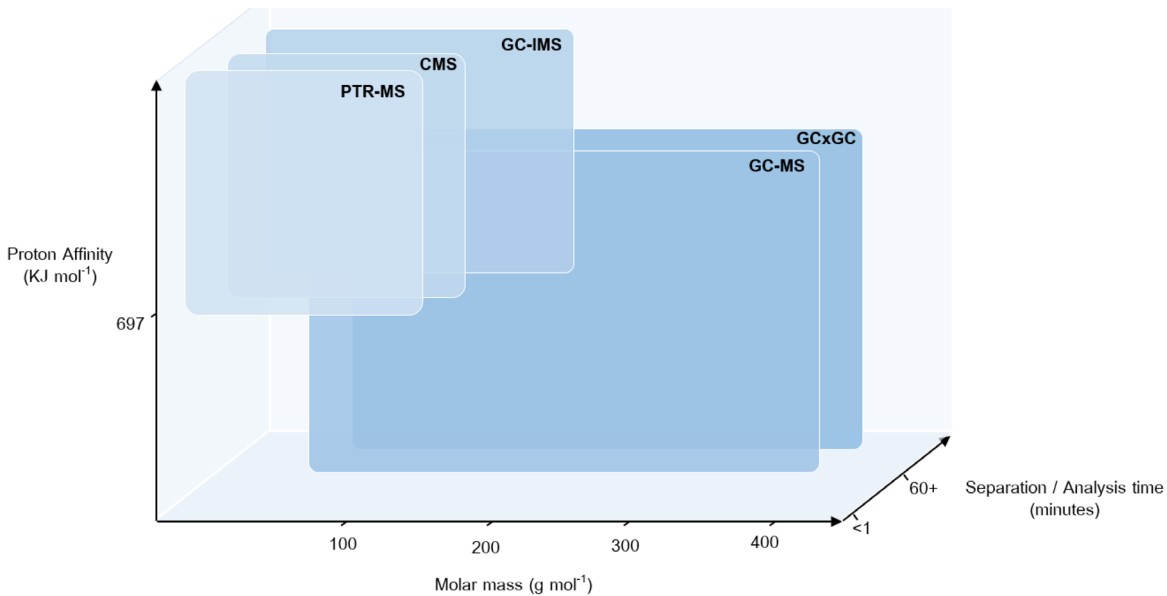

**Figure 4** Multi-instrument use in breath sampling. Operational space of the analytical technologies used in EMBER for the analysis of volatile organic compounds in exhaled breath, including proton transfer reaction mass spectrometry (PTR-MS), atmospheric pressure chemical ionisation-mass spectrometry (CMS), gas chromatography ion mobility spectrometry (GC-IMS), gas chromatography mass spectrometry (GC-MS) and two-dimensional gas chromatography-mass spectrometry (GC×GC-MS). Comparing the typical molar mass range detectable; selectivity in detection owing to the type of ionisation involved and the proton affinity of analytes; and the inclusion of a chromatographic separation affecting total time of analysis. The online technologies involving chemical ionisation (PTR-MS, CMS and GC-IMS) can be used in-clinic owing to short analysis times but only detect lower molar mass molecules with a proton affinity higher than 697 KJ/mol. Offline chromatographic techniques (GC-MS and GC×GC-MS) detect a wider range of compounds independent of proton affinity; however, the techniques have longer analysis times and involve sample transportation and storage. EMBER, East Midlands Breathomics Pathology Node.

identifiers and subsequently pseudonymised prior to integration.

An informatics pipeline will be created to facilitate the transfer of chemometric data from remote computers to the data repository. This will include tools to: (1) enforce the correct labelling of data sets (eg, study number, visit and type/source of sample) prior to automated validated transfer to the repository; (2) record information about the sample process; and (3) search and extract data sets from the repository for subsequent analysis. Prior to analysis, clinical and chemometric data will be integrated using the study number and any potentially identifiable information will be removed.

## BREATH PROFILING
The technologies used in the VOC study during discovery and replication phases are:
Offline technologies:
► ReCIVA+GC-MS.
► ReCIVA+GC×x GC-MS.
Online technologies:
► GC-IMS.
► PTR-MS.
► APCI-MS.

Offline technologies will underpin the discovery analyses owing to their ability to identify chemical identity and their recognition as the analytical gold standard in exhaled breath VOC analysis.[53]

In contrast, online technologies will be used for VOC biomarker replication and at the recovery visits owing to their portability and potential for future point of care testing (figure 4). A brief description of the core VOC platforms is provided below.

A CE-marked breath sampling device (ReCIVA) developed by Owlstone Medical will be used to sample breath onto two adsorbent Tenax tubes. Participants will be asked to breathe through the ReCIVA face mask for a maximum of 900 seconds, aiming for collection of ≥80% of the target sample volume of 1 L, after which the Tenax tubes will be transferred to the laboratory for analysis. This effectively allows decoupling of the breath sampling from the breath sensor and analysis platforms in selected patients that are not able to mobilise to a real-time breath sampling device. The Owlstone ReCIVA sampler will be used in breath collection for offline technologies namely GC-MS and GC×GC-MS. The ReCIVA sampler is capable of entraining oxygen and is therefore suitable for patients with mild respiratory failure requiring low flow rates of oxygen to maintain target oxygen saturations.[48]

### Gas chromatography and mass spectroscopy
GC-MS is a commonly applied methodology used to accurately measure trace gases in complex mixtures such as exhaled air.[53] Preconcentrating breath volatiles by various means and subsequent analysis constitute a reliable and sensitive method for VOC analysis.[54] Despite its high sensitivity, it is, however, a time-consuming technique

and carries a risk of contamination at the preconcentration step. It is also not suitable for online and multiple measurements limiting its use as a point-of-care testing technology for VOC.[55]

The instrument used will be an Agilent 7890A gas chromatogram with a 5977a quadrupole mass spectrometer (Agilent Technologies Ltd, Stockport, UK), interfaced with a Markes Unity 2 thermal desorptionunit (Markes International Ltd, Llantrisant, UK).

## Comprehensive two-dimensional gas chromatography-mass spectrometry (GC×GC-MS)

GC×GC-MS is an advanced analytical technique for the analysis of complex organic matrices; its main advantage is the unparalleled separation power it affords over conventional one-dimensional chromatographic techniques.[56] Previous research, although sparse, has demonstrated the potential of GC×GC-MS for breath analysis with the number of VOC detected exceeding those detected by conventional GC-MS.[57 58] GC×GC-MS of breath metabolites has been used for the identification of biomarkers related to glucose metabolism,[59 60] tuberculosis[61] and radiation response.[62] This has generated interest within the breath research community; however, such studies were conducted on a small scale (<50 patients) and involved the use of expensive detectors and modulators. Method development and analysis of the data-rich GC×GC chromatograms, however, can be time-consuming and require specialist knowledge.

The instrument used will be an Agilent 7890A gas chromatogram, fitted with a G3486A CFT flow modulator and a three-way splitter plate coupled to a flame ionisation detector and a HES 5977B quadrupole mass spectrometer (Agilent Technologies Ltd, Stockport, UK), interfaced with a Markes TD-100xr thermal desorption autosampler (Markes International Ltd, Llantrisant, UK).

Proton Transfer Reaction Time-of-Flight Mass Spectromery (PTR-ToF-MS) is a real-time technique, capable of simultaneously measuring the evolution of multiple gas metabolites from a single breath. It has been used for the identification of potential useful VOC biomarkers for diagnosis of a variety of diseases including various cancers,[63–65] liver disease[66 67] and respiratory disease.[68] It has several advantages in clinical settings, such as the speed of sampling, the instant result achieved and the lack of need for sample storage or shipping. However, owing to the lack of preconcentration or chromatographic separation, sensitivity and definitive compound identification can be somewhat limited when compared with GC-MS.

Two breath sampling devices will be used. The first device is a Loccioni SOFIA GSI-S; the subject is required to exhale a single breath, five times (three if providing five samples proves too difficult) into a sterile mouthpiece connected to an electrostatic bacterial/viral filter while wearing a nose clip (all CE marked). Flow from the mouthpiece passes into a gas sampling interface capnograph (Loccioni GSI-S – CE marked), and real-time user feedback of flow is provided on screen, allowing the

regulation of the breath sampling rate. The gas sampling interface acts to simultaneously trigger the acquisition of the Proton Transfer Reaction Time-of-Flight Mass Spectromery (PTR-ToF-MS) data and the exhaled breath travels through the capnograph down a heated sample line into the ion source of the PTR-ToF-MS.

The second breath sampling device is a ReCIVA breath sampler (Owlstone) with one of the adsorbent Tenax tubes replaced with an outlet tube adapted for online sampling. The exhaled breath is transferred to the PTR-ToF-MS via a heated transfer line connected to the outlet tube, continuously drawn at a constant flow rate by the PTR-ToF-MS. The online adaptation of the consumable adsorbent tube does not affect the CE mark of the ReCIVA sampling device.

Once the breath sample reaches the PTR-ToF-MS, via either breath sampler, the breath mixes with protonated water ($H_3O^+$) inducing proton transfer to the target VOCs present, resulting in their ionisation. Sample ions are then guided into the time of flight mass spectrometer, and mass spectra, showing the abundance and mass of the VOCs present, collected throughout the exhalation. Following sampling, mouthpieces, filters and nose clips are disposed of, and all patient-contacted surfaces were wiped down with antiseptic cleaning wipes in preparation for the next patient.

The instrument used will be a Kore Series II high performance proton transfer reaction time of flight-mass spectrometer (Kore Technology Ltd, Cambridge, UK).

## Gas chromatography - Ion Mobility Spectrometry (GC-IMS) (B&S Analytiks)

GC-IMS allows the detection of VOCs down to ultratrace level (µg/L – to pg/L – range). For years, IMS has been used to discover potential discriminatory breath VOC in lung cancer,[69 70] chronic obstructive pulmonary disease (COPD)[71 72] and asthma.[72] Sampling takes place using a Spiroscout spirometer. The patients exhale through a disposable mouth piece connected to a Teflon tube. A piezoelectric pressure sensor is used to monitor the breathing profile; this opens the sampling valve at the appropriate point in the breath profile to collect end-tidal breath in a sample loop of 10 mL volume. After filling this loop, the collected sample air is then transferred to a multicapillary column for a chromatographic separation, which is achieved in 12 min. The separated molecules are then transferred into the IMS, ionised and then separated according to their mobility in a weak electric field.

The technology's multiple advantages of ultrasensitivity, portability, online sampling and short analysis time (typical analysis time of 10 min) with real-time detection brings a promise to provide immediate and potentially reliable results for point of care breath diagnostics. Another concept with IMS devices is that once the required breath signatures have been discovered using GC-MS, IMS offers the potential to be 'tuned' for selective detection of VOC.

The instrument used will be a BioScout a multicapillary column GC-IMS with a $^{63}$Ni ion source, interfaced with

a SpiroScout breath sampler (BS Analytik, Dortmund, Germany).

## APCI-MS semiportable compact version (Advion)

APCI-MS is one of less sensitive but more affordable versions of mass spectrometers released to the commercial market in recent years. The device uses APCI to produce ions. Although the most common use of APCI-MS systems is the detection in liquid chromatography applications, the technique has proven to be a valuable tool for direct measurement of VOC in air,[73 74] food[75 76] and breath.[77 78] Recently, the technique has shown potential for online, real-time profiling of pseudometabolites in exhaled breath[79] with sensitivity comparable with other techniques. By combining miniaturised mass spectrometry technology with APCI techniques, adequate quality of on-site, real-time measurements with minimal or no sample preparation requirement can be provided. This is a desirable outcome as it overcomes main limitation of using standard breath analysis method in clinical setting, which is a need for breath sample collection followed by desorption and time-consuming laboratory analysis.

Preconcentrating breath gas by various means and subsequent analysis by means of GC-MS constitute a reliable and sensitive set of methods for VOCs analysis.

There remains an overall lack of standardisation and rigour across these technologies that hindered previous advancements in breath discovery; something we intend to minimise.

The instrument used will be an Advion Compact Mass Spectrometer Express, with atmospheric pressure chemical ionisation, interfaced with a heated breath sampling line (Advion, New York, USA).

## CHEMOMETRIC PROCESSING AND DATA ANALYSIS

GC-MS breath data will be aligned, deconvoluted and the features for each participant will be extracted. The extracted features will be grouped and classified by retention index and mass spectrum. The registered and aligned data will be linked to participant metadata to generate a breath matrix. Data handling and analysis will be performed by a senior statistician.

The breath matrix is a × matrix where n is the number of subjects and p is the number of VOC. The breath matrix is high dimensional with ≫ and many potentially correlated VOC. In view of this, we will employ sparse partial least squares discriminant analysis[80] to investigate which of the VOC can identify breathlessness. We will also investigate which of the VOC can discriminate between the different disease states including acute exacerbations of asthma and COPD and pneumonia. In addition to the supervised methods, unsupervised methods will be explored, specifically sparse principle component analysis.[81]

Extracted VOC will also be investigated. Relationships between VOC and patient-reported acute breathlessness will be analysed using logistic regression model. VOC associated with patient reported acute breathlessness will be incorporated into multinomial logistic regression models in conjunction with CRP, BNP, blood eosinophils and troponin-I, pathology biomarkers currently in use for diagnosing undifferentiated breathlessness. In addition to the conventional binary and multinomial logistic regression models.[82]

## ETHICS AND DISSEMINATION

Publications will be prepared according to the MRC-EMBER consortium agreement and the University of Leicester publications policy. All intended publications will be submitted to the EMBER executive board for review and comments within 60 days of journal submission. Authorship will be according to contribution and internationally recognised guidance on journal authorship.

### Patient and public involvement

A series of consultations have taken place with our patient involvement team within the NIHR Biomedical Research Centre (Respiratory Theme) and across the wider BRC patient and public involvement group. Representations from the paediatrics team were also present. This group was sent copies of the participant documentation for review and discussion. Various revisions have been made following on from these discussions.

**Author affiliations**
[1]Department of Respiratory Sciences, College of Life Sciences, University of Leicester, Leicester, UK
[2]University Hospitals of Leicester NHS Trust, Leicester, UK
[3]Department of Chemistry, University of Leicester, Leicester, UK
[4]Department of Chemistry, Loughborough University, Loughborough, UK
[5]NIHR Leicester Clinical Research Facility, Leicester, UK
[6]Paediatric Clinical Investigation Centre, Leicester, UK
[7]Department of Cardiovascular Sciences, Cardiovascular Research Centre, University of Leicester, Leicester, UK
[8]Leicester NIHR Biomedical Research Centre (Cardiovascular Theme), Leicester, UK
[9]Discovery Medicine, Respiratory Therapeutic Area, GlaxoSmithKline PLC, Stevenage, UK
[10]Department of Cardiovascular Sciences, University of Leicester, Leicester, UK

**Contributors** SS, CEB, NG, PT and PSM conceived the study, obtained funding, wrote the study protocol, obtained ethical and Medicines and Healthcare products Regulatory Agency (MHRA) approvals for the study and coordinated the deployment of analytical testing methods for breath analysis. WI took the lead in writing the manuscript with support from SS. Planning and recruitment of adult participants was carried out by WI, SJo, BPa, AAw, RPh, GFo, AYo, RR and CWh. Paediatrics study design was conceived by EG and CB and participants recruited by TM and CF. Analytical chemistry team formed of MW, RC, DS, DR and LB expertly handled all the breath samples and planned an analysis structure. MR, a senior statistician, constructed a statistics and data analysis plan in conjunction with SS. Bioinformatics pipeline and electronic CRFs developed by RF and BZ. All authors contributed to the study design and study protocol.

**Funding** This research was funded by the Medical Research Council (MRC), *Engineering and Physical Sciences Research Council* (*EPSRC*) Stratified Medicine Grant for Molecular Pathology Nodes (Grant No. MR/N005880/1) and Midlands Asthma and Allergy Research Association (MAARA), and carried out at the University Hospitals of Leicester NHS Trust and University of Leicester, supported by the NIHR Leicester Biomedical Research Centre and the NIHR Leicester Clinical Research Facility. The views expressed are those of the author(s) and not necessarily those of the NHS, the NIHR or the Department of Health and Social Care. The authors would like to acknowledge the invaluable efforts of the research nurses responsible for

the in-clinic sample collection as well as the input from the wider East Midlands Breathomics Pathology Node consortium (members list can be found at: https://ember.le.ac.uk/web).

**Disclaimer** The views expressed are those of the author(s) and not necessarily those of the NHS and NIHR or the Department of Health.

**Competing interests** SS has performed advisory services for Owlstone Medical.

**Patient consent for publication** Not required.

**Ethics approval** The study has obtained full ethical approval from the London South East Research ethics Committee, REC reference 16/LO/1747.

**Provenance and peer review** Not commissioned; externally peer reviewed.

**Open access** This is an open access article distributed in accordance with the Creative Commons Attribution 4.0 Unported (CC BY 4.0) license, which permits others to copy, redistribute, remix, transform and build upon this work for any purpose, provided the original work is properly cited, a link to the licence is given, and indication of whether changes were made. See: https://creativecommons.org/licenses/by/4.0/.

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
