## [Reviewer comments · BMJ Open]

ARTICLE DETAILS

TITLE (PROVISIONAL)	Assessment of Breath Volatile Organic Compounds in Acute Cardio-respiratory Breathlessness: A protocol describing a Prospective Real World Observational Study
AUTHORS	Ibrahim, Wadah; Wilde, Michael; Cordell, Rebecca; Salman, Dahlia; Ruszkiewicz, Dorota; Bryant, Luke; Richardson, Matthew; Free, Robert; Zhao, Bo; Yousuf, Ahmed; White, Christobelle; Russell, Richard; Jones, Sheila; Patel, Bharti; Awal, Asia; Phillips, Rachael; Fowkes, Graham; McNally, Teresa; Foxon, Clare; Bhatt, Hetan; Peltrini, Rosa; Singapuri, Amisha; Hargadon, Beverley; Suzuki, T; Ng, Leong; Gaillard, Erol; Beardsmore, Caroline; Ryanna, Kimuli; Pandya, Hitesh; Coates, Tim; Monks, Paul; Greening, Neil; Brightling, Christopher; Thomas, Paul; Siddiqui, Salman

VERSION 1 – REVIEW

REVIEWER	Dr Annette Dent Director, Respiratory Sciences, Department of Thoracic Medicine, The Prince Charles Hospital, Respiratory Researcher, The University of Queensland Thoracic Research Centre, QLD, Australia
REVIEW RETURNED	20-Aug-2018

GENERAL COMMENTS	Thank you for the opportunity to review this proposal. The proposed study shows great potential in discriminating subjects with an unknown cause of breathlessness by their exhaled breath VOC profile. The use of multiple types of analysers, both on-line and off-line, should provide detailed information that will permit the authors to define and replicate the VOC profile in the subjects. I have a few questions and suggestions regarding the proposal. Firstly, I would suggest that the references may need to be updated and extended, in particular for VOC studies in COPD and asthma. There have been many studies performed in both of these areas in recent years. In 2.4 Discovery and replication studies: 2.41 and 2.42 need more clarification and more detail. As it reads at the moment 2.42 is almost exactly the same as 2.4.1 and I am unable to come to the same total of subjects to be recruited. In the discovery and replication studies one of the subject groups are to be age matched healthy volunteers. Could the definition of this group be further explained eg are they lifetime non-smokers, no major medical conditions, etc. There is no detail or method described for the replication method. 2.54 VOC sampling table 2 describes the baseline and follow-up assessment tests to be performed on the different subjects groups. An explanation should be provided as to why the PTRMS is only to be used for the recovery phase of subjects with COPD, asthma, pneumonia and heart failure while the GC-IMS is used for the baseline measurements for the same subject groups. With regards
--

	to the lung function measurements, though FOT is a simple measurement to perform it is still not as well recognised as spirometry and I would suggest that the study would benefit by also measuring spirometry. Spirometry will provide classification of the severity of the asthma and COPD in the subjects. As a simple measurement to perform, I would have expected that FOT would have been used in the paediatric group as well. Is there a reason why this has not been included for this patient group? It may be useful to also discriminate between COPD patients admitted with a bacterial infection or a viral infection as early diagnosis will improve treatment options. This could be an ideal use of the VOC in the admission of a patient with an exacerbation. Is this something the authors have considered? 2.5.6 Physiological characterisation - echo measurements were described as past tense, should these be described in future tense? 2.5.7 Recovery follow-up - is this by patient identification or are other methods to be used to identify the subject as in recovery? In some of the groups there will be frequent admissions, how will post-recovery be defined in this study? 3. Breath profiling - the methods to be used for collecting and analysing the breath samples have not been fully described. The equipment has been generically described but not identified as to their brand, model and company that produces them. Could the authors provide a detailed description of how the breath is to be collected and analysed on each of the instruments. There is insufficient detail provided. The measurement of ambient VOCs may also be useful so as to be subtracted from the subjects VOC profile. Will background VOCs from food or smoking be minimised by a withholding period? The authors have described one of the limitations of the study is prior acute treatment exposure. How will this be addressed. Statistics in this field is very complex and not my area of expertise. Do the authors have the assistance of a statistician for this study?
--	---

REVIEWER	Paul Brinkman Amsterdam UMC, Location AMC, Department of Respiratory Medicine, Amsterdam, The Netherlands.
REVIEW RETURNED	26-Aug-2018

GENERAL COMMENTS	In this manuscript Ibrahim et al. describe a study protocol for a real world, prospective, observational study across three admission units. The proposed study will seek to use both offline and online technologies for sampling and analysing exhaled volatile organic compounds (VOCs) for the purpose of identifying common conditions that present with acute cardio-respiratory breathlessness. While the protocol is clear and straight forward, a few topics might benefit from some additional information. - Will enrolment only be considered after final diagnosis (which is by protocol allowed to take a maximum of 24 hours)? Or will a patient be recruited based on screening outcomes (as soon as possible after presentation) and in case of unclear diagnosis within 24 hours be excluded again? - Can the researchers expand on the planned strategy for the offline sampling? What are the settings of the ReCIVA? How many desorption tubes are sampled? What is the planned sampling volume? Will the breath sampling be based on tidal volumes or e.g. full capacity?
--

	- Will the Breath VOC sampling through the different devices be based on a random order, or will the GC-IMS always be the final test? - Considering a patient needs to perform at least four VOCs sampling/analysis tests (ReCIVA, PTR-MS, GC-IMS, APCI-MS) + FOT + FENO, is this feasible for a patients suffering from acute breathlessness? - Is the potential of 'drop-outs' a risk that should be taken into account as it concerns the power calculation? - In the paragraph concerning GC-MS breath data analysis through sPLS-DA, the researchers note the high dimensionality of the n*p matrix. Would 'overfitting' or the influence of therapy be a potential risk here?
--	---

VERSION 1 – AUTHOR RESPONSE

Reviewer A:

C7: I would suggest that the references may need to be updated and extended, in particular for VOC studies in COPD and asthma. There have been many studies performed in both of these areas in recent years.

R7: Many thanks for raising this point. Recent studies have now been included. Please refer to section1, page 8 -9 of the revised manuscript.

C8: In 2.4 Discovery and replication studies: 2.41 and 2.42 need more clarification and more detail. As it reads at the moment 2.4.2 is almost exactly the same as 2.4.1 and I am unable to come to the same total of subjects to be recruited.

R8: Thank you for pointing this out. The manuscript has now been corrected to reflect this. In addition, a table has been generated for ease of reference – see below. Please refer to page 14-15, table 1, section 2.4.1 and 2.4.2 of the revised version.

Disease Category	Discovery	Replication
Acute Adult Asthma	55	55
Acute COPD	55	55
Acute Heart Failure	55	55
Community Acquired Pneumonia	55	55
Adult healthy volunteers	55	55
Acute paediatrics Asthma	50	25
Paediatrics healthy volunteers	50	25
Total sample	375	325

C9: Could the definition of healthy volunteers be further explained e.g. are they lifetime non-smokers, no major medical conditions, etc.

R9: For the purposes of this study, healthy volunteers are defined as participants who have no prior history of any of the pre-selected primary diagnoses of interest (asthma, COPD and heart failure) and

have not been admitted to hospital with community acquired pneumonia within 6 weeks of the baseline study visit. This has now been updated. Please refer to page 13, section 2.4, and line 362 of the revised manuscript.

C10: 2.54: An explanation should be provided as to why the PTRMS is only to be used for the recovery phase of subjects with COPD, asthma, pneumonia and heart failure while the GC-IMS is used for the baseline measurements for the same subject groups

R10: Gas chromatography mass spectrometry (GC-MS), currently the gold standard technique for detecting VOC in breath, is what we've adopted towards our primary end point. PTRMS is a next generation technology that characteristically measures low molecular weight compounds and although forms an important component in our study, it is an exploratory rather than core study technology. Furthermore, it was not possible to deploy PTRMS device in the allocated acute clinical setting. We have now clarified this on the tracked version, please refer to page 16 of the revised manuscript, section 2.5.4, line 427.

C11: FOT: It's a simple measurement to perform but is still not as well recognised as spirometry and I would suggest that the study would benefit by also measuring spirometry.

R11: Thank you for raising this important point. Our aim was to deploy a lung function technique that can be used by all but not some. We realised early on, following the feasibility study, that spirometry coverage will be poor as it is effort dependant and most of our patients are unwell with moderate to severe disease exacerbation. Following on from our patient consultation, we adopted the hand held FOT which proved popular as it's considerably easier to perform. We accept the reviewer's point that spirometry is important, however, its importance lies in the stable rather than the acute disease, research on spirometry use in acute disease is largely non trivial.

C12: Is there a reason why FOT has not been included for paediatrics?

R12: Given the nature of our study and the vulnerable cohort of unwell children we're recruiting, it was decided following ethics review, to restrict the assessments to the bare minimum to limit the patient burden.

C13: It may be useful to also discriminate between COPD patients admitted with a bacterial infection or a viral infection as early diagnosis will improve treatment options. This could be an ideal use of the VOC in the admission of a patient with an exacerbation. Is this something the authors have considered?

R13: Thank you, this has been considered. Blood and sputum based biomarkers are deployed in all which will help clinically stratify patients by causality, lacking specific bacterial or viral identification. Further discussions are currently taking place to expand the study in a specific subset of patients to involve sputum microbiome and virome testing which can later be linked with discovered VOCs.

C14: 2.5.6 Physiological characterisation - echo measurements were described as past tense, should these be described in future tense?

R14: This grammatical error has now been corrected. Please refer to page 18, section 2.5.61, line 456.

C15: 2.5.7 Recovery follow-up - is this by patient identification or are other methods to be used to

identify the subject as in recovery? In some of the groups there will be frequent admissions, how will post-recovery be defined in this study?

R15: Patient recovery will be defined as

(i) Patient reported recovery from the acute exacerbation spell and back to their baseline extended MRC score or clinician defined recovery from the acute exacerbation spell
and

(ii) At least 6 weeks post exacerbation event (up to 6 months).

Patients that re admit to hospital between visits 1 and 2, can have additional visit 1 assessments. Visit 2 will be taken as recovery following the subsequent admission. If a patient is admitted to hospital after visit 2 then they will be eligible to be recruited as a new study participant. This has clarified on page 19, section 2.5.7 of the revised manuscript.

C16: 3. Breath profiling - the methods to be used for collecting and analysing the breath samples have not been fully described. The equipment has been generically described but not identified as to their brand, model and company that produces them. Could the authors provide a detailed description of how the breath is to be collected and analysed on each of the instruments. The measurement of ambient VOCs may also be useful so as to be subtracted from the subjects VOC profile. Will background VOCs from food or smoking be minimised by a withholding period?

R16: Many thanks for raising this point. A detailed description has been added to each of the technologies. Please refer to page 21, section 3, line 533 for ReCIVA, page 22, section 3.3 for PTR-MS, page 23, section 3.4 for GC-IMS.

No withholding period will be applied prior to breath sampling, however, a detailed questionnaire of food and environmental exposures will need to be completed by each participant at the point of testing.

C17: The authors have described one of the limitations of the study is prior acute treatment exposure. How will this be addressed?

R17: Thank you for raising this important point. This is a pragmatic well powered acute study where multivariate analysis will account for relevant exposures. These will be identified by reviewing clinical admission notes.

C18: Statistics in this field is very complex and not my area of expertise. Do the authors have the assistance of a statistician for this study?

R18: There is an analytical team including a senior statistician that will expertly handle and analyse our data. This has now been made clear on section 4 of the revised version under chemometric analysis and data processing, page 24, line 654.

Reviewer B:

C19: Will enrolment only be considered after final diagnosis (which is by protocol allowed to take a maximum of 24 hours)? Or will a patient be recruited based on screening outcomes (as soon as possible after presentation) and in case of unclear diagnosis within 24 hours be excluded again?

R19: Participants are only recruited after final diagnosis is confirmed. As per local trust policy, all patients are reviewed by a senior decision maker within 24 hours of admission.

In an effort to reduce data variability and minimise bias, an independent panel consisting of two senior acute clinicians will review all clinical notes and determine the primary diagnosis of highest probability.

Please refer to page 19, section 2.6 on clinical adjudication on the revised manuscript.

C20: Can the researchers expand on the planned strategy for the offline sampling? What are the settings of the ReCIVA? How many desorption tubes are sampled? What is the planned sampling volume? Will the breath sampling be based on tidal volumes or e.g. full capacity?

R20: Thank you for raising this point. A CE marked breath sampling device (ReCIVA®) developed by Owlstone Medical, will be used to sample breath onto two adsorbent Tenax tubes. Participants will be asked to breathe through the ReCIVA face mask for a maximum of 300 seconds, aiming for collection of $\geq 80\%$ of the target sample volume of 1 litre, after which the Tenax tubes will be transferred to the laboratory for analysis. We have now updated the manuscript to accurately describe this. Please refer to page 21, section3, line 534.

C21: Will the Breath VOC sampling through the different devices be based on a random order, or will the GC-IMS always be the final test?

R21: ReCIVA testing, which provides samples for GC-MS and GCxGC-MS is always prioritised as it determines our primary end point. Patients are then tested on the other online and offline technologies based on their clinical state and degree of breathlessness.

C22: Considering a patient needs to perform at least four VOCs sampling/analysis tests (ReCIVA, PTR-MS, GC-IMS, APCI-MS) + FOT + FENO, is this feasible for patients suffering from acute breathlessness?

R22: All recruited participants are assessed by the trial clinician to ensure patient safety prior to testing. We have set a maximum of 30 minutes to be spent with each patient at any given time to avoid any potential distress caused by testing. Participants are encouraged to report any testing related discomfort or concerns to the research team to terminate the sampling process. Clinically unstable patients and those dependant on high flow oxygen will automatically be excluded at the screening stage. This has now been added to the revised version of the manuscript. Please refer to page 19, section 2.5.6, line 463.

C23: Is the potential of 'drop-outs' a risk that should be taken into account as it concerns the power calculation?

R23: Thank you for bringing this up. This will not be the case as the power calculation was based on the first visit.

C24: In the paragraph concerning GC-MS breath data analysis through sPLS-DA, the researchers note the high dimensionality of the $n \times p$ matrix. Would 'overfitting' or the influence of therapy be a potential risk here?

R24: Careful conduction of the analysis will reduce the risk of over-fitting. The sPLS-DA is used to tackle the high dimensionality of the matrix, p the number of variables is much greater than n the number of subjects. Sparse PCA would be another way of addressing the high dimensionality. There are always issues with high dimensional data, but the sparse methods are a reasonable approach to avoiding gross over-fitting. PCA is a data reduction method, even if after doing sparse PCA/DA there are still a fair few variables of interest then further regression methods will be considered later down the line that will accommodate this situation.

VERSION 2 – REVIEW

REVIEWER	Dr Annette Dent Director, Respiratory Sciences, The Prince Charles Hospital, Australia
-----------------	--

REVIEW RETURNED	03-Nov-2018
-------------

GENERAL COMMENTS	Thank you for the revised manuscript. Your responses has answered most of my concerns however it would benefit from some further amendments. Also, I was unable to complete the review because on p 24 part of the manuscript was illegible. My comments refer to the rest of the manuscript include the following:  1. The definition of a healthy control subject should be tightened. Not having prior history of the conditions of interest may not be sufficient as other major diseases may have a VOC profile as does smokers, etc. 2. In your response you stated that FOT will not be performed on the paediatric subjects however in table 3, Schedule of Assessments, it appears that the crosses indicating the performance of FOT in the paediatric subjects were placed on the row for lung function tests (line 44). Could this please be clarified. Also, there is no description of the equipment and methods to be used for the FOT test or the FENO. Normally, the FOT test takes less effort than a FENO measurement and therefore would be more suitable in paediatric subjects. 3. It is mentioned that a MRC score may be used as a measure of breathlessness. This would be a useful tool to be used on all subjects for a comparison between samples collected during acute breathlessness and in recovery. 4. As the second reviewer mentioned, the number of breath samples to be collected seems to be extreme in an acute setting. The allowance of 30 minutes to perform all of the tests does not seem sufficient particularly when it includes 5 minutes for each ReCIVA sample to be collected. 5. There are still past tense used in the description of the echo measurements p 20, lines 430, 432 and present tense on 434 instead of future tense. 6. The recovery follow up section states that if a subject is admitted to hospital after visit 2 they will be eligible to be recruited as a new study participant. Would this skew the results unless you are planning to look at a subset of subjects that have been readmitted and perform an intrasubject comparison. 7. The description of the equipment to be used is still incomplete. There is no details for the brand, model and manufacturer for GC-MS, GCxGC-MS, PTRMS, APCI-MS, Spiroscout spirometer. 8. Why are two breath sampling methods, ReCIVA and Loccioni SOFIA GSI-S to be used in the PTRMS analyser? Does this add to the study outcomes?
---

REVIEWER	Paul Brinkman Amsterdam UMC, University of Amsterdam Department of Respiratory Medicine Amsterdam, The Netherlands
REVIEW RETURNED	21-Oct-2018

GENERAL COMMENTS	I would like to thank the authors for their extensive response and have no further questions. Looking forward to the outcomes of the interesting study.
---

VERSION 2 – AUTHOR RESPONSE

Comment 1: The definition of a healthy control subject should be tightened. Not having prior history of the conditions of interest may not be sufficient as other major diseases may have a VOC profile as

does smokers, etc

Response 1: For the intended purpose of our real world pragmatic study and in an attempt to maximise generalisability and applicability, we chose to include ex-smokers with no respiratory function limitation or a diagnosis of COPD as part of our healthy cohorts. We ruled out current smokers and ex-smokers with defined cardio-respiratory illnesses. As we are still in the discovery phase, it would be interesting to detect VOCs that are specific to subgroups of healthy volunteers based on their comorbidities and/or smoking status. Further sub analysis of healthy volunteers will be considered accordingly.

Comment 2: In your response you stated that FOT will not be performed on the paediatric subjects however in table 3, Schedule of Assessments, it appears that the crosses indicating the performance of FOT in the paediatric subjects were placed on the row for lung function tests (line 44). Could this please be clarified? Also, there is no description of the equipment and methods to be used for the FOT test or the FENO. Normally, the FOT test takes less effort than a FENO measurement and therefore would be more suitable in paediatric subjects.

R2: Description and make of the equipment used have now been added to the manuscript, please refer to section 2.5.6. The choice of testing in the paediatric cohort was driven by the patient and public involvement group (PPI), as paediatrics is only a sub-study of our main trial, it was decided early on that FOT is not be performed to minimise testing burden, table 3 has now been amended to reflect this.

Comment 3: It is mentioned that a MRC score may be used as a measure of breathlessness. This would be a useful tool to be used on all subjects for a comparison between samples collected during acute breathlessness and in recovery

R3: Extended MRC score (eMRC) will be used as a measure of breathlessness in all participants (acute patients and healthy volunteers) at the initial visit and later in recovery, this will be used for sample comparison between visits. For the purpose of our study, breathlessness is defined as a 1 unit increase above patient reported baseline in the eMRC score. This has now been clarified further in section 2.5.1, line 385.

Comment 4: As the second reviewer mentioned, the number of breath samples to be collected seems to be extreme in an acute setting. The allowance of 30 minutes to perform all of the tests does not seem sufficient particularly when it includes 5 minutes for each ReCIVA sample to be collected.

R4: Time and motion analysis was completed and 30 minutes were found to be sufficient for completion of breath testing. ReCIVA takes average of 8 minutes to complete, PTRMS: 2-3 minutes, APCI-CMS: 10-15 seconds, GC-IMS: 12 seconds and Oscillometry: 16 seconds. The research testing room is part of the admission ward with almost no time lost transporting patients. The instruments are usually pre-set and lined up in the testing room and the team works harmoniously to facilitate a seamless flow through testing stations. The research nurses are accompanied by an experienced clinical research fellow who assesses the acuity of each recruited participant and if the testing is felt to be burdensome, it is broken down further to shorter intervals or discontinued.

Comment 5: There are still past tense used in the description of the echo measurements p 20, lines 430, 432 and present tense on 434 instead of future tense.

R5: This has now been amended.

Comment 6: The recovery follow up section states that if a subject is admitted to hospital after visit 2 they will be eligible to be recruited as a new study participant. Would this skew the results unless you

are planning to look at a subset of subjects that have been readmitted and perform an intrasubject comparison?

R6: The percentage of the re-admitted patients recruited to our study is expected to be very low (2-5%). Readmission data will be analysed separately and will not be included in our total study numbers. Further analysis of this data, including intra-subject comparability, will be considered.

Comment 7: The description of the equipment to be used is still incomplete. There is no details for the brand, model and manufacturer for GC-MS, GCxGC-MS, PTRMS, APCI-MS, Spiroscout spirometer.
R7: This has now been updated, please refer to section 3 of the manuscript.

Comment 8: Why are two breath sampling methods, ReCIVA and Loccioni SOFIA GSI-S to be used in the PTRMS analyser? Does this add to the study outcomes?

R8: The Loccioni sampler is a single incentivised breath sample, where the patient is asked to breath out once, maintaining an exhalation at a set pressure, similar to a forced exhaled but not as difficult or standardised. The ReCIVA in PTRMS is used for tidal breath sampling. Papers report breath manoeuvre can affect composition so we're collecting both to determine if one long breath or tidal breathing is better for detecting VOCs in PTRMS. This paper highlights how different sampling techniques can impact results of breath analysis, which shaped the basis of our consideration. (Impact of sampling procedures on the results of breath analysis Wolfram, Miekisch et al 2008 J. Breath Res. 2 026007)

VERSION 3 – REVIEW

REVIEWER	Annette Dent, Director Respiratory Sciences The Prince Charles Hospital, Queensland, Australia
REVIEW RETURNED	09-Dec-2018
GENERAL COMMENTS	I am satisfied with the revised manuscript.